# Unsupervised Deep Structure Learning by Recursive Dependency Analysis

## Abstract

We introduce an unsupervised structure learning algorithm for deep, feed-forward, neural networks. We propose a new interpretation for depth and inter-layer connectivity where a hierarchy of independencies in the input distribution is encoded in the network structure. This results in structures allowing neurons to connect to neurons in any deeper layer skipping intermediate layers. Moreover, neurons in deeper layers encode low-order (small condition sets) independencies and have a wide scope of the input, whereas neurons in the first layers encode higher-order (larger condition sets) independencies and have a narrower scope. Thus, the depth of the network is automatically determined—equal to the maximal order of independence in the input distribution, which is the recursion-depth of the algorithm. The proposed algorithm constructs two main graphical models: 1) a generative latent graph (a deep belief network) learned from data and 2) a deep discriminative graph constructed from the generative latent graph. We prove that conditional dependencies between the nodes in the learned generative latent graph are preserved in the class-conditional discriminative graph. Finally, a deep neural network structure is constructed based on the discriminative graph. We demonstrate on image classification benchmarks that the algorithm replaces the deepest layers (convolutional and dense layers) of common convolutional networks, achieving high classification accuracy, while constructing significantly smaller structures. The proposed structure learning algorithm requires a small computational cost and runs efficiently on a standard desktop CPU.

## 1 Introduction

Over the last decade, deep neural networks have proven their effectiveness in solving many challenging problems in various domains such as speech recognition (Graves & Schmidhuber, 2005), computer vision (Krizhevsky et al., 2012; Girshick et al., 2014; Simonyan & Zisserman, 2014; Szegedy et al., 2015) and machine translation (Collobert et al., 2011b). As compute resources became more available, large scale models having millions of parameters could be trained on massive volumes of data, to achieve state-of-the-art solutions for these high dimensionality problems. Building these models requires various design choices such as network topology, cost function, optimization technique, and the configuration of related hyper-parameters.

In this paper, we focus on the design of network topology—structure learning. Generally, exploration of this design space is a time consuming iterative process that requires close supervision by a human expert. Many studies provide guidelines for design choices such as network depth (Simonyan & Zisserman, 2014), layer width (Zagoruyko & Komodakis, 2016), building blocks (Szegedy et al., 2015), and connectivity (He et al., 2016; Huang et al., 2016). Based on these guidelines, these studies propose several meta-architectures, trained on huge volumes of data. These were applied to other tasks by leveraging the representational power of their convolutional layers and fine-tuning their deepest layers for the task at hand (Donahue et al., 2014; Hinton et al., 2015; Long et al., 2015; Chen et al., 2015; Liu et al., 2015). However, these meta-architecture may be unnecessarily large and require large computational power and memory for training and inference.

The problem of model structure learning has been widely researched for many years in the probabilistic graphical models domain. Specifically, Bayesian networks for density estimation and causal discovery (Pearl, 2009; Spirtes et al., 2000). Two main approaches were studied: score-based

(search-and-score) and constraint-based. Score-based approaches combine a scoring function, such as BDe (Cooper & Herskovits, 1992) and BIC (Ripley, 2007), with a strategy for searching through the space of structures, such as greedy equivalence search (Chickering, 2002). Adams et al. (2010) introduced an algorithm for sampling deep belief networks (generative model) and demonstrated its applicability to high-dimensional image datasets.

Constraint-based approaches (Pearl, 2009; Spirtes et al., 2000) find the optimal structures in the large sample limit by testing conditional independence (CI) between pairs of variables. They are generally faster than score-based approaches (Yehezkel & Lerner, 2009) and have a well-defined stopping criterion (e.g., maximal order of conditional independence). However, these methods are sensitive to errors in the independence tests, especially in the case of high-order conditional-independence tests and small training sets.

Motivated by these methods, we propose a new interpretation for depth and inter-layer connectivity in deep neural networks. We derive a structure learning algorithm such that a hierarchy of independencies in the input distribution is encoded in the network structure, where the first layers encode higher-order independencies than deeper layers. Thus, the number of layers is automatically determined. Moreover, a neuron in a layer is allowed to connect to neurons in deeper layers skipping intermediate layers. An example of a learned structure, for MNIST, is given in Figure 1.

We describe our recursive algorithm in two steps. In Section 2 we describe a base case—a single-layer structure learning. In Section 3 we describe multi-layer structure learning by applying the key concepts of the base case, recursively (proofs are provided in Appendix A). In Section 4 we discuss related work. We provide experimental results in Section 5, and conclude in Section 6.

**Preliminaries**. Consider $\boldsymbol{X} = \{X_i\}_{i=1}^N$ a set of observed (input) random variables, $\boldsymbol{H} = \{H_j\}_{j=1}^K$ a set of latent variables, and $Y$ a class variable. Our algorithm constructs three graphical models and an auxiliary graph. Each variable is represented by a single node and a single edge may connect two distinct nodes. Graph $\mathcal{G}$ is a generative DAG defined over the observed and latent variables $\boldsymbol{X} \cup \boldsymbol{H}$. Graph $\mathcal{G}_{\mathrm{Inv}}$ is called a stochastic inverse of $\mathcal{G}$. Graph $\mathcal{G}_{\mathrm{D}}$ is a discriminative model defined over the observed, latent, and class variables $\boldsymbol{X} \cup \boldsymbol{H} \cup Y$. An auxiliary graph $\mathcal{G}_X$ is defined over $\boldsymbol{X}$ (a CPDAG; an equivalence class of a Bayesian network) and is generated and maintained as an internal state of the algorithm. The parents set of a node $X$ in $\mathcal{G}$ is denoted $\boldsymbol{Pa}(X; \mathcal{G})$. The order of an independence relation is defined to be the condition set size. For example, if $X_1$ and $X_2$ are independent given $X_3$ and $X_4$, denoted $X_1 \perp\!\!\!\perp X_2 | \{X_3, X_4\}$, then the independence order is two.

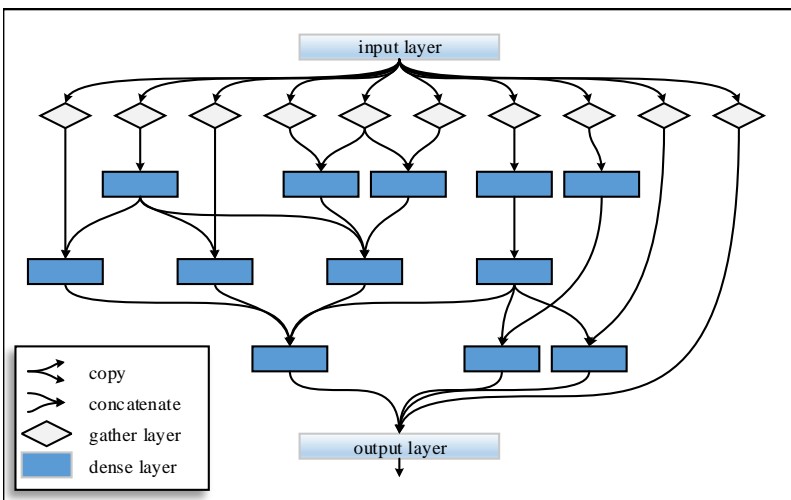

Figure 1: An example of a structure learned by our algorithm (classifying MNIST digits). Neurons in a layer may connect to neurons in any deeper layer. Depth is determined automatically. Each gather layer selects a subset of the input, where each input variable is gathered only once. A neural route, starting with a gather layer, passes through densely connected layers where it may split (copy) and merge (concatenate) with other routes in correspondence with the hierarchy of independencies identified by the algorithm. All routes merge into the final output layer (e.g., a softmax layer).

## 2  SINGLE LAYER STRUCTURE LEARNING

We start by describing the key concepts of our approach using a simple scenario: learning the connectivity of a single-layer neural network.

### 2.1  CONSTRUCTING A GENERATIVE GRAPH

Assume the input joint distribution $p(\boldsymbol{X})$ complies with the following property.

**Assumption 1.** *The joint distribution $p(\boldsymbol{X})$ is faithful to a DAG $\mathcal{G}$ over observed $\boldsymbol{X}$ and latent nodes $\boldsymbol{H}$, where for all $X \in \boldsymbol{X}$ and $H \in \boldsymbol{H}$, $\boldsymbol{Pa}(X;\mathcal{G}) \subseteq \boldsymbol{H}$ and $\boldsymbol{Pa}(H;\mathcal{G}) \subseteq \boldsymbol{H} \backslash H$.*

$$p\Big(\boldsymbol{X};\mathcal{G}\Big) = \int p\Big(\boldsymbol{X},\boldsymbol{H};\mathcal{G}\Big)d\boldsymbol{H} = \int p\Big(\boldsymbol{H}\Big)\prod_{i=1}^{N}p\Big(X_i\Big|\boldsymbol{Pa}(X_i;\mathcal{G})\Big)d\boldsymbol{H}. \qquad (1)$$

Note that the generative graphical model $\mathcal{G}$ can be described as a layered deep belief network where parents of a node in layer $m$ can be in any deeper layer, indexes greater than $m$, and not restricted to the next layer $m+1$. This differs from the common definition of deep belief networks (Hinton et al., 2006; Adams et al., 2010) where the parents are restricted to layer $m+1$.

It is desired to learn an efficient graph $\mathcal{G}$ having small sets of parents and a simple factorization of $p(\boldsymbol{H})$ while maintaining high expressive power. We first construct an auxiliary graph, a CPDAG (Spirtes et al., 2000), $\mathcal{G}_X$ over $\boldsymbol{X}$ (an equivalence class of a fully visible Bayesian network) encoding only *marginal* independencies[1] (empty condition sets) and then construct $\mathcal{G}$ such that it can mimic $\mathcal{G}_X$ over $\boldsymbol{X}$, denoted $\mathcal{G}_X \preceq \mathcal{G}$ (Pearl, 2009). That is, preserving all conditional dependencies of $\boldsymbol{X}$ in $\mathcal{G}_X$.

The simplest connected DAG that encodes statistical independence is the v-structure, a structure with three nodes $X_1 \to X_3 \leftarrow X_2$ in which $X_1$ and $X_2$ are marginally independent $X_1 \perp\!\!\!\perp X_2$ and conditionally dependent $X_1 \not\perp\!\!\!\perp X_2|X_3$. In graphs encoding only marginal independencies, dependent nodes form a clique. We follow the procedure described by Yehezkel & Lerner (2009) and decompose $\boldsymbol{X}$ into autonomous sets (complying with the Markov property) where one set, denoted $\boldsymbol{X}_{\mathrm{D}}$ (descendants), is the common child of all other sets, denoted $\boldsymbol{X}_{\mathrm{A}1},\ldots,\boldsymbol{X}_{\mathrm{A}K}$ (ancestor sets). We select $\boldsymbol{X}_{\mathrm{D}}$ to be the set of nodes that have the lowest topological order in $\mathcal{G}_X$. Then, by removing $\boldsymbol{X}_{\mathrm{D}}$ from $\mathcal{G}_X$ (temporarily for this step), the resulting $K$ disjoint sets of nodes (corresponding to $K$ disjoint substructures) form the $K$ ancestor sets $\{\boldsymbol{X}_{\mathrm{A}i}\}_{i=1}^{K}$. See an example in Figure 2.

Next, $\mathcal{G}$ is initialized to an empty graph over $\boldsymbol{X}$. Then, for each ancestor set $\boldsymbol{X}_{\mathrm{A}i}$ a latent variable $H_i$ is introduced and assigned to be a common parent of the pair $(\boldsymbol{X}_{\mathrm{A}i},\boldsymbol{X}_{\mathrm{D}})$. Thus,

$$p\Big(\boldsymbol{X};\mathcal{G}\Big) = \int \prod_{i=1}^{K}\left[p\Big(H_i\Big)\prod_{X \in \boldsymbol{X}_{\mathrm{A}i}}p\Big(X\Big|H_i\Big)\right]\prod_{X' \in \boldsymbol{X}_{\mathrm{D}}}p\Big(X'\Big|\boldsymbol{H}\Big)d\boldsymbol{H}. \qquad (2)$$

Note that the parents of two ancestor sets are distinct, whereas the parents set of the descendant set is composed of all the latent variables.

In the auxiliary graph $\mathcal{G}_X$, for each of the resulting v-structures $(\boldsymbol{X}_{\mathrm{A}i} \to \boldsymbol{X}_{\mathrm{D}} \leftarrow \boldsymbol{X}_{\mathrm{A}j})$, a link between a parent and a child can be replaced by a common latent parent without introducing new independencies. For example, in Figure 2-[b], $\boldsymbol{X}_{\mathrm{A}1} = \{A\}$, $\boldsymbol{X}_{\mathrm{A}2} = \{B\}$, and $\boldsymbol{X}_{\mathrm{D}} = \{C,D,E\}$. Adding a common latent parent (Figure 3-[a]) $H_A$ (or $H_B$) and removing all the edges from $\boldsymbol{X}_{\mathrm{A}1}$ (or $\boldsymbol{X}_{\mathrm{A}2}$) to $\boldsymbol{X}_{\mathrm{D}}$ preserves the conditional dependence $A \not\perp\!\!\!\perp B|\{C,D,E\}$.

Algorithm 1 summarizes the procedure of constructing $\mathcal{G}$ having a single latent layer. Note that we do not claim to identify the presence of confounders and their inter-relations as in Elidan et al. (2001); Silva et al. (2006); Asbeh & Lerner (2016). Instead, we augment a fully observed Bayesian network with latent variables, while preserving conditional dependence.

---

[1]In Section 3, conditional independencies are considered, the construction of $\mathcal{G}_X$ and $\mathcal{G}$ is interleaved, and the ability of $\mathcal{G}$ to mimic $\mathcal{G}_X$ over $\boldsymbol{X}$ is described recursively.

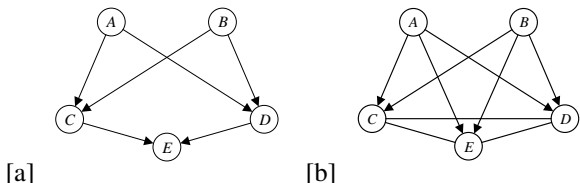

[a]         [b]

Figure 2: [a] An example of a Bayesian network encoding ground-truth conditional independencies (a DAG underlying observed data) and [b] a corresponding CPDAG ($\mathcal{G}_X$) constructed by testing only marginal independencies. Only $A$ and $B$ are marginally independent (d-separated in [a], $A \perp\!\!\!\perp B$), where $C$ and $D$ are marginally dependent ($C \not\perp\!\!\!\perp D$) and therefore connected in [b]. Thus, nodes $\{C, D, E\}$, having the lowest topological order, form a descendant set $\boldsymbol{X}_{\mathrm{D}} = \{C, D, E\}$ and nodes $A$ and $B$ form two distinct ancestor sets, $\boldsymbol{X}_{\mathrm{A}1} = \{A\}, \boldsymbol{X}_{\mathrm{A}2} = \{B\}$—disjoint if $\{C, D, E\}$ is removed from the graph.

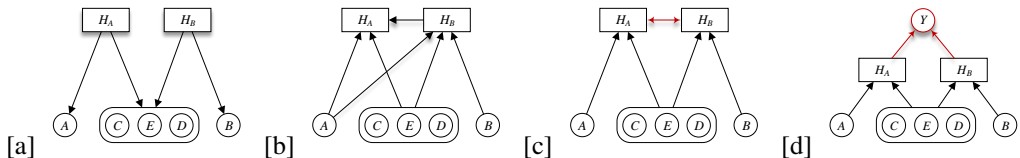

[a]        [b]        [c]        [d]

Figure 3: [a] An example of a graph $\mathcal{G}$ (corresponding to $\mathcal{G}_X$ in Figure 2-[b]). [b] A stochastic inverse generated by the algorithm presented by Stuhlmüller et al. (2013). [c] A stochastic inverse generated by our method where the graph is a projection of a latent structure. A dependency induced by a latent $Q$ is described using a bi-directional edge $H_A \leftrightarrow H_B$. [d] A discriminative structure $\mathcal{G}_{\mathrm{D}}$ having a class node $Y$ that provides an explaining away relation for $H_A \leftrightarrow H_B$. That is, the latent $Q$ is replaced by an observed common child $Y$.

## 2.2 Constructing a Stochastic Inverse

It is important to note that $\mathcal{G}$ represents a generative distribution of $\boldsymbol{X}$ and is constructed in an unsupervised manner (class variable $Y$ is ignored). Hence, we construct $\mathcal{G}_{\mathrm{Inv}}$, a graphical model that preserves all conditional dependencies in $\mathcal{G}$ but has a different node ordering in which the observed variables, $\boldsymbol{X}$, have the highest topological order (parentless)—a stochastic inverse of $\mathcal{G}$. Note that conditional dependencies among $\boldsymbol{X}$ are not required to be preserved in the stochastic inverse as these are treated (simultaneously) as observed variables (highest topological order).

Stuhlmüller et al. (2013); Paige & Wood (2016) presented a heuristic algorithm for constructing such stochastic inverses where the structure is a DAG (an example is given in Figure 3-[b]). However, these DAGs, though preserving all conditional dependencies, may omit many independencies and add new edges between layers.

We avoid limiting $\mathcal{G}_{\mathrm{Inv}}$ to a DAG and instead limit it to be a projection of another latent structure (Pearl, 2009). That is, we assume the presence of *additional* hidden variables $\boldsymbol{Q}$ that are not in $\mathcal{G}_{\mathrm{Inv}}$ but induce dependency[2] among $\boldsymbol{H}$. For clarity, we omit these variables from the graph and use bi-directional edges to represent the dependency induced by them. An example is given in Figure 3-[c] where a bi-directional edge represents the effect of some variable $Q \in \mathbf{Q}$ on $H_A$ and $H_B$. We construct $\mathcal{G}_{\mathrm{Inv}}$ in two steps:

1. Invert all $\mathcal{G}$ edges (invert inter-layer connectivity).

2. Connect each pair of latent variables, sharing a common child in $\mathcal{G}$, with a bi-directional edge.

This simple procedure ensures $\mathcal{G} \preceq \mathcal{G}_{\mathrm{Inv}}$ over $\boldsymbol{X} \cup \boldsymbol{H}$ while maintaining the exact same number of edges between the layers (Proposition 1, Appendix A).

---

[2]For example, "interactive forks" (Pearl, 2009).

---

**Algorithm 1:** Marginal Connectivity Learning

---

**Input:** $\boldsymbol{X}$: observed nodes, and `Indep`: an oracle for testing statistical independence.
**Output:** $\mathcal{G}$, a latent structure over $\boldsymbol{X}$ and $\boldsymbol{H}$

1   *initialize $\mathcal{G}_X \longleftarrow$ a complete graph over $\boldsymbol{X}$*

2   **begin**

3      **foreach** *pair of connected nodes $X_i, X_j$ in $\mathcal{G}_X$ if* `Indep`$(X_i, X_j)$   ▷ find independencies

4      **do**

5         disconnect $X_i$ and $X_j$

6         direct edges $X_i \rightarrow X_c \leftarrow X_j$ for every common neighbor $X_c$

7      $\boldsymbol{X}_\mathrm{D} \longleftarrow$ nodes having the lowest topological order        ▷ identify autonomous sets

8      $\{\boldsymbol{X}_{\mathrm{A}i}\}_{i=1}^K \longleftarrow$ disjoint sets, after removing $\boldsymbol{X}_\mathrm{D}$ from $\mathcal{G}_X$

9      $\mathcal{G} \longleftarrow$ an empty graph over $\boldsymbol{X}$

10      add $K$ latent variables $\boldsymbol{H} = \{H_i\}_{i=1}^K$ to $\mathcal{G}$        ▷ create a latent layer

11      set each $H_i$ to be a parent of $\{\boldsymbol{X}_{\mathrm{A}1} \cup \boldsymbol{X}_\mathrm{D}\}$        ▷ connect

12      **return** $\mathcal{G}$

---

## 2.3   Constructing a Discriminative Graph

Recall that $\mathcal{G}$ encodes the generative distribution of $\boldsymbol{X}$ and $\mathcal{G}_\mathrm{Inv}$ is the stochastic inverse. We further construct a discriminative graph $\mathcal{G}_\mathrm{D}$ by replacing bi-directional dependency relations in $\mathcal{G}_\mathrm{Inv}$, induced by $\boldsymbol{Q}$, with explaining-away relations by adding the observed class variable $Y$. Node $Y$ is set in $\mathcal{G}_\mathrm{D}$ to be the common child of the leaves in $\mathcal{G}_\mathrm{Inv}$ (latents introduced after testing marginal independencies) (see an example in Figure 3-[d]). This preserves the conditional dependency relations of $\mathcal{G}_\mathrm{Inv}$. That is, $\mathcal{G}_\mathrm{D}$ can mimic $\mathcal{G}_\mathrm{Inv}$ over $\boldsymbol{X}$ and $\boldsymbol{H}$ given $Y$ (Proposition 2, Appendix A). It is interesting to note that the generative and discriminative graphs share the exact same inter-layer connectivity (inverted edge-directions). Moreover, introducing node $Y$ provides an "explaining away" relation between latents, uniquely for the classification task at hand.

## 2.4   Constructing a Feed-Forward Neural Network

We construct a neural network based on the connectivity in $\mathcal{G}_\mathrm{D}$. Sigmoid belief networks (Neal, 1992) have been shown to be powerful neural network density estimators (Larochelle & Murray, 2011; Germain et al., 2015). In these networks, conditional probabilities are defined as logistic regressors. Similarly, for $\mathcal{G}_\mathrm{D}$ we may define for each latent variable $H' \in \boldsymbol{H}$,

$$p(H' = 1|\boldsymbol{X}') = \mathrm{sigm}\left(\boldsymbol{W}'\boldsymbol{X}' + b'\right) \tag{3}$$

where $\mathrm{sigm}(x) = 1/(1 + \exp(-x))$, $\boldsymbol{X}' = \boldsymbol{Pa}(H'; \mathcal{G}_\mathrm{D})$, and $(\boldsymbol{W}', b')$ are the parameters of the neural network. Nair & Hinton (2010) proposed replacing each binary stochastic node $H'$ by an infinite number of copies having the same weights but with decreasing bias offsets by one. They showed that this infinite set can be approximated by

$$\sum_{i=1}^N \mathrm{sigm}(v - i + 0.5) \approx \log(1 + e^v), \tag{4}$$

where $v = \boldsymbol{W}'\boldsymbol{X}' + b'$. They further approximate this function by $\max(0, v + \epsilon)$ where $\epsilon$ is a zero-centered Gaussian noise. Following these approximations, they provide an approximate probabilistic interpretation for the ReLU function, $\max(0, v)$. As demonstrated by Jarrett et al. (2009) and Nair & Hinton (2010), these units are able to learn better features for object classification in images.

In order to further increase the representational power, we represent each $H'$ by a set of neurons having ReLU activation functions. That is, each latent variable $H'$ in $\mathcal{G}_\mathrm{D}$ is represented in the neural network by a dense (fully-connected) layer. Finally, the class node $Y$ is represented by a softmax layer.

## 3 RECURSIVE MULTI-LAYER STRUCTURE LEARNING

We now extend the method of learning the connectivity of a single layer into a method of learning multi-layered structures. The key idea is to recursively introduce a new and deeper latent layer by testing $n$-th order conditional independence ($n$ is the condition set size) and connect it to latent layers created by previous recursive calls that tested conditional independence of order $n + 1$. The method is described in Algorithm 2. It is important to note that conditional independence is tested only between input variables $\boldsymbol{X}$ and condition sets do not include latent variables. Conditioning on latent variables or testing independence between them is not required as the algorithm adds these latent variables in a specific manner, preserving conditional dependencies between the input variables.

---

**Algorithm 2:** Recursive Latent Structure Learning (multi-layer)

1   **RecurLatStruct** $(\mathcal{G}_X, \boldsymbol{X}, \boldsymbol{X}_{\mathrm{ex}}, n)$
     **Input:** an initial DAG $\mathcal{G}_X$ over observed $\boldsymbol{X}$ & exogenous nodes $\boldsymbol{X}_{\mathrm{ex}}$ and a desired resolution $n$.
     **Output:** $\mathcal{G}$, a latent structure over $\boldsymbol{X}$ and $\boldsymbol{H}$

2      **if** *the maximal indegree of $\mathcal{G}_X(\boldsymbol{X})$ is below $n + 1$* **then**             ▷ exit condition
3         $\mathcal{G} \longleftarrow$ an observed layer $\boldsymbol{X}$
4         **return** $\mathcal{G}$

5      $\mathcal{G}'_X \longleftarrow$ IncreaseResolution $(\mathcal{G}_X, n)$          ▷ $n$-th order independencies
6      $\{\boldsymbol{X}_{\mathrm{D}}, \boldsymbol{X}_{\mathrm{A}1}, \ldots, \boldsymbol{X}_{\mathrm{A}K}\} \longleftarrow$ SplitAutonomous $(\boldsymbol{X}, \mathcal{G}'_X)$     ▷ identify autonomies
7      **for** $i \in \{1 \ldots K\}$ **do**
8         $\mathcal{G}_{\mathrm{A}i} \longleftarrow$ RecurLatStruct $(\mathcal{G}'_X, \boldsymbol{X}_{\mathrm{A}i}, \boldsymbol{X}_{\mathrm{ex}}, n + 1)$        ▷ a recursive call
9      $\mathcal{G}_{\mathrm{D}} \longleftarrow$ RecurLatStruct $(\mathcal{G}'_X, \boldsymbol{X}_{\mathrm{D}}, \boldsymbol{X}_{\mathrm{ex}} \cup \{\boldsymbol{X}_{\mathrm{A}i}\}_{i=1}^{K}, n + 1)$     ▷ a recursive call
10     $\mathcal{G} \longleftarrow$ Group $(\mathcal{G}_{\mathrm{D}}, \mathcal{G}_{\mathrm{A}1}, \ldots, \mathcal{G}_{\mathrm{A}K})$            ▷ merge results
11     create latent variables $\boldsymbol{H}^{(n)} = \{H_1^{(n)}, \ldots, H_K^{(n)}\}$ in $\mathcal{G}$      ▷ create a latent layer
12     set each $H_i^{(n)}$ to be a parent of $\{\boldsymbol{H}_{\mathrm{A}i}^{(n+1)} \cup \boldsymbol{H}_{\mathrm{D}}^{(n+1)}\}$          ▷ connect
13     where $\boldsymbol{H}_{\mathrm{A}i}^{(n+1)}$ and $\boldsymbol{H}_{\mathrm{D}}^{(n+1)}$ are the sets of parentless latents in $\mathcal{G}_{\mathrm{A}i}$ and $\mathcal{G}_{\mathrm{D}}$, respectively.
14     **return** $\mathcal{G}$

---

The algorithm maintains and recursively updates an auxiliary graph $\mathcal{G}_X$ (a CPDAG) over $\boldsymbol{X}$ and utilizes it to construct $\mathcal{G}$. Yehezkel & Lerner (2009) introduced an efficient algorithm (RAI) for constructing a CPDAG over $\boldsymbol{X}$ by a recursive application of conditional independence tests with increasing condition set sizes ($n$). Our algorithm is based on this framework for updating the auxiliary graph $\mathcal{G}_X$ (Algorithm 2, lines 5 and 6).

The algorithm starts with $n = 0$, $\mathcal{G}_X$ a complete graph, and a set of exogenous nodes $\boldsymbol{X}_{\mathrm{ex}} = \emptyset$. The set $\boldsymbol{X}_{\mathrm{ex}}$ is exogenous to $\mathcal{G}_X$ and consists of parents of $\boldsymbol{X}$.

The function IncreaseResolution (Algorithm 2-line 5) disconnects (in $\mathcal{G}_X$) conditionally independent variables in two steps. First, it tests dependency between $\boldsymbol{X}_{\mathrm{ex}}$ and $\boldsymbol{X}$, i.e., $X \perp\!\!\!\perp X' | \boldsymbol{S}$ for every connected pair $X \in \boldsymbol{X}$ and $X' \in \boldsymbol{X}_{\mathrm{ex}}$ given a condition set $\boldsymbol{S} \subset \{\boldsymbol{X}_{\mathrm{ex}} \cup \boldsymbol{X}\}$ of size $n$. Next, it tests dependency within $\boldsymbol{X}$, i.e., $X_i \perp\!\!\!\perp X_j | \boldsymbol{S}$ for every connected pair $X_i, X_j \in \boldsymbol{X}$ given a condition set $\boldsymbol{S} \subset \{\boldsymbol{X}_{\mathrm{ex}} \cup \boldsymbol{X}\}$ of size $n$. After removing the corresponding edges, the remaining edges are directed by applying two rules (Pearl, 2009; Spirtes et al., 2000). First, v-structures are identified and directed. Then, edges are continually directed, by avoiding the creation of new v-structures and directed cycles, until no more edges can be directed. Following the terminology of Yehezkel & Lerner (2009), we say that this function increases the graph d-separation resolution from $n - 1$ to $n$.

The function SplitAutonomous (Algorithm 2-line 6) identifies autonomous sets in a graph in two steps, as described in Algorithm 1 lines 7 and 8. An autonomous set in $\mathcal{G}_X$ includes all its nodes' parents (complying with the Markov property) and therefore a corresponding latent structure can be constructed independently using a recursive call. Thus, the algorithm is recursively and independently called for the ancestor sets (Algorithm 2 lines 7–8), and then called for the descendant set while treating the ancestor sets as exogenous (Algorithm 2 line 9).

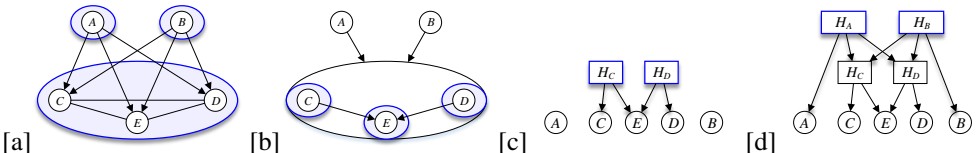

Figure 4: An example trace of Algorithm 2 where $p(\boldsymbol{X})$ is faithful to the DAG in Figure 3-[a]. [a] a CPDAG encoding only marginal independencies ($n = 0$) and the identified autonomous sub-structures (line 6 in the algorithm). [b] A CPDAG $\mathcal{G}_X$ over $\{C, D, E\}$ encoding conditional independencies up to second order $n = 2$ (nodes $A$ and $B$ are exogenous). [c] Graph $\mathcal{G}$ is created for the autonomous set $\{C, D, E\}$ by introducing latents $\{H_C, H_D\}$. At $n = 2$ nodes $C$ and $D$ are identified as autonomous ancestors and $E$ as an autonomous descendant. [d] Graph $\mathcal{G}$ is created at $n = 0$ for all nodes $\{A, B, H_C, H_D\}$ by introducing $\{H_A, H_B\}$, where $\{H_C, H_D\}$ represent the autonomous subset $\{C, D, E\}$.

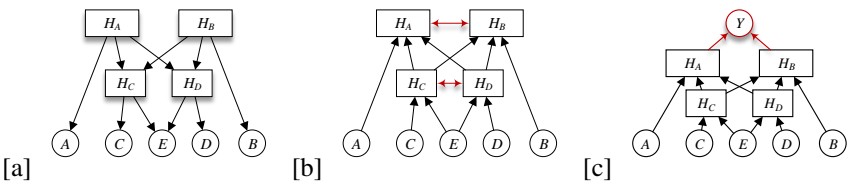

Figure 5: An example of [a] a structure $\mathcal{G}$ (the corresponding auxiliary graph $\mathcal{G}_X$ is depicted in Figure 2-[a]), [b] its stochastic inverse $\mathcal{G}_{\mathrm{Inv}}$ described as a projection of another latent structure, and [c] a corresponding discriminative model obtained by expressing dependency relations among latents (bi-directional edges) with an "explaining away" relation induced by a class node.

Each recursive call returns a latent structure for each autonomous set. Recall that each latent structure encodes a generative distribution over the observed variables where layer $\boldsymbol{H}^{(n+1)}$, the last added layer (parentless nodes), is a representation of the input $\boldsymbol{X'} \subset \boldsymbol{X}$. By considering only layer $\boldsymbol{H}^{(n+1)}$ of each latent structure, we have the same simple scenario discussed in Section 2—learning the connectivity between $\boldsymbol{H}^{(n)}$, a new latent layer, and $\boldsymbol{H}^{(n+1)}$, treated as an "input" layer. Thus, latent variables are introduced as parents of the $\boldsymbol{H}^{(n+1)}$ layers, as described in Algorithm 2 lines 11–13. A simplified example is given in Figure 4.

Next, a stochastic inverse $\mathcal{G}_{\mathrm{Inv}}$ is constructed as described in Section 2—all the edge directions are inverted and bi-directional edges are added between every pair of latents sharing a common child in $\mathcal{G}$. An example graph $\mathcal{G}$ and a corresponding stochastic inverse $\mathcal{G}_{\mathrm{Inv}}$ are given in Figure 5. A discriminative structure $\mathcal{G}_{\mathrm{D}}$ is then constructed by removing all the bi-directional edges and adding the class node $Y$ as a common child of layer $\boldsymbol{H}^{(0)}$, the last latent layer that is added (Figure 5-[c]). Finally, a neural network is constructed based on the connectivity of $\mathcal{G}_{\mathrm{D}}$. That is, each latent node, $H \in \boldsymbol{H}^{(n)}$, is replaced by a set of neurons, and each edge between two latents, $H \in \boldsymbol{H}^{(n)}$ and $H' \in \boldsymbol{H}^{(n+1)}$, is replaced by a bipartite graph connecting the neurons corresponding to $H$ and $H'$.

## 4 RELATED WORK

Recent studies have focused on automating the exploration of the design space, posing it as a hyper-parameter optimization problem and proposing various approaches to solve it. (Miconi, 2016) learns the topology of an RNN network introducing structural parameters into the model and optimize them along with the model weights by the common gradient descent methods. Smith et al. (2016) takes a similar approach incorporating the structure learning into the parameter learning scheme, gradually growing the network up to a maximum size.

A common approach is to define the design space in a way that enables a feasible exploration process and design an effective method for exploring it. Zoph & Le (2016) (NAS) first define a set of hyper-parameters characterizing a layer (number of filters, kernel size, stride). Then they use a

controller-RNN for finding the optimal sequence of layer configurations for a "trainee network". This is done using policy gradients (REINFORCE) for optimizing the objective function that is based on the accuracy achieved by the "trainee" on a validation set. Although this work demonstrates capabilities to solve large-scale problems (Imagenet), it comes with huge computational cost. In a following work, Zoph et al. (2017) address the same problem but apply a hierarchical approach. They use NAS to design network modules on a small-scale dataset (CIFAR-10) and transfer this knowledge to a large-scale problem by learning the optimal topology composed of these modules. Baker et al. (2016) use reinforcement learning as well and apply Q-learning with epsilon-greedy exploration strategy and experience replay. Negrinho & Gordon (2017) propose a language that allows a human expert to compactly represent a complex search-space over architectures and hyper-parameters as a tree and then use methods such as MCTS or SMBO to traverse this tree. Smithson et al. (2016) present a multi objective design space exploration, taking into account not only the classification accuracy but also the computational cost. In order to reduce the cost involved in evaluating the network's accuracy, they train a Response Surface Model that predicts the accuracy at much lower cost, reducing the number of candidates that go through actual validation accuracy evaluation. Another common approach for architecture search is based on evolutionary strategies to define and search the design space. (Real et al., 2017; Miikkulainen et al., 2017) use evolutionary algorithm to evolve an initial model or blueprint based on its validation performance.

Common to all these recent studies is the fact that structure learning is done in a supervised manner, eventually learning a discriminative model. Moreoever, these approaches require huge compute resources, rendering the solution unfeasible for most applications given limited compute and time resources.

## 5 EXPERIMENTS

We evaluate the quality of the learned structure in two experiments:

- Classification accuracy as a function of network depth and size for a structure learned directly from MNIST pixels.
- Classification accuracy as a function of network size on a range of benchmarks and compared to common topologies.

All the experiments were repeated five times where average and standard deviation of the classification accuracy were recorded. In all of our experiments, we used a ReLU function for activation, ADAM (Kingma & Ba, 2015) for optimization, and applied batch normalization (Ioffe & Szegedy, 2015) followed by dropout (Srivastava et al., 2014) to all the dense layers. All optimization hyper-parameters that were tuned for the vanilla topologies were also used, without additional tuning, for the learned structures. For the learned structures, all layers were allocated an equal number of neurons. Threshold for independence tests, and the number of neurons-per-layer were selected by using a validation set. Only test-set accuracy is reported.

Our structure learning algorithm was implemented using the Bayesian network toolbox (Murphy, 2001) and Matlab. We used Torch7 (Collobert et al., 2011a) and Keras (Chollet, 2015) with the TensorFlow (Abadi et al., 2015) back-end for optimizing the parameters of both the vanilla and learned structures.

### 5.1 NETWORK DEPTH, NUMBER OF PARAMETERS, AND ACCURACY

We analyze the accuracy of structures learned by our algorithm as a function of the number of layers and parameters. Although network depth is automatically determined by the algorithm, it is implicitly controlled by the threshold used to test conditional independence (partial-correlation test in our experiments). For example, a high threshold may cause detection of many independencies leading to early termination of the algorithm and a shallow network (a low threshold has the opposite effect). Thus, four different networks having 2, 3, 4, and 5 layers, using four different thresholds, are learned for MNIST. We also select three configurations of network sizes: a baseline (normalized to 1.00), and two configurations in which the number of parameters is 0.5, and 0.375 of the baseline network (equal number of neurons are allocated for each layer).

Classification accuracies are summarized in Table 1. When the number of neurons-per-layers is large enough (100%) a 3-layer network achieves the highest classification accuracy of 99.07% (standard deviation is 0.01) where a 2-layer dense network has only a slight degradation in accuracy, 99.04%. For comparison, networks with 2 and 3 fully connected layers (structure is not learned) with similar number of parameters achieve 98.4% and 98.75%, respectively. This demonstrates the efficiency of our algorithm when learning a structure having a small number of layers. In addition, for a smaller neuron allocation (50%), deeper structures learned by our algorithm have higher accuracy than shallower ones. However, a decrease in the neurons-per-layer allocation has a greater impact on accuracy for deeper structures.

| num. of paramters | 2 layers | 3 layers | 4 layers | 5 layers |
|---|---|---|---|---|
| 1.00 | 99.04 (0.01) | 99.07 (0.012) | 99.07 (0.035) | 99.07 (0.052) |
| 0.5 | 98.96 (0.03) | 98.98 (0.035) | 99.02 (0.04) | 99.02 (0.05) |
| 0.375 | 98.96 (0.035) | 98.94 (0.04) | 98.93 (0.041) | 98.93 (0.049) |

Table 1: Mean classification accuracy [%] (and standard deviation) of structures learned from MNIST images as a function of network depth and number of parameters (normalized). For comparison, when a structure is not learned, networks with 2 and 3 dense layers, achieve 98.4% and 98.75% accuracy, respectively (having the same size as learned structures at configuration "100%").

## 5.2 LEARNING THE STRUCTURE OF THE DEEPEST LAYERS IN COMMON TOPOLOGIES

We evaluate the quality of learned structures using five image classification benchmarks. We compare the learned structures to common topologies (and simpler hand-crafted structures), which we call "vanilla topologies", with respect to network size and classification accuracy. The benchmarks and vanilla topologies are described in Table 2. In preliminary experiments we found that, for SVHN and ImageNet, a small subset of the training data is sufficient for learning the structure (larger training set did not improve classification accuracy). As a result, for SVHN only the basic training data is used (without the extra data), i.e., 13% of the available training data, and for ImageNet 5% of the training data is used. Parameters were optimized using all of the training data.

| benchmark | vanilla topology | | |
|---|---|---|---|
| dataset | topology | description | size |
| MNIST (LeCun et al., 1998) | None
MNIST-Man | learn a structure directly from pixels
32-64-FC:128 | 
127K |
| SVHN (Netzer et al., 2011) | Maxout NiN
SVHN-Man | (Chang & Chen, 2015)
16-16-32-32-64-FC:256 | 1.6M
105K |
| CIFAR 10 (Krizhevsky & Hinton, 2009) | VGG-16
WRN-40-4 | (Simonyan & Zisserman, 2014)
(Zagoruyko & Komodakis, 2016) | 15M
9M |
| CIFAR 100 (Krizhevsky & Hinton, 2009) | VGG-16 | (Simonyan & Zisserman, 2014) | 15M |
| ImageNet (Deng et al., 2009) | AlexNet | (Krizhevsky et al., 2012) | 61M |

Table 2: Benchmarks and vanilla topologies. MNIST-Man and SVHN-Man topologies were manually created by us. MNIST-Man has two convolutional layer (32 and 64 filters each) and one dense layer with 128 neurons. SVHN-Man was created as a small network reference having reasonable accuracy compared to Maxout-NiN. In the first row we indicate that in one experiment a structure for MNIST was learned from the pixels and feature extracting convolutional layers were not used.

Convolutional layers are powerful feature extractors for images exploiting domain knowledge, such as spatial smoothness, translational invariance, and symmetry. We therefore evaluate our algorithm by using the first convolutional layers of the vanilla topologies as "feature extractors" (mostly below 50% of the vanilla network size) and learning a deep structure from their output. That is, the deepest layers of the vanilla network (mostly over 50% of the network size) is removed and replaced by a structure learned by our algorithm in an unsupervised manner. Finally, a softmax layer is added and the entire network parameters are optimized.

First, we demonstrate the effect of replacing a different amount of the deepest layers and the ability of the learned structure to replace feature extraction layers. Table 3 describes classification accuracy achieved by replacing a different amount of the deepest layers in VGG-16. For example, column "conv.10" represents learning a structure using the activations of conv.10 layer. Accuracy and the normalized number of network parameters are reported for the overall network, e.g., up to conv.10 + the learned structure. Column "vanilla" is the accuracy achieved by the VGG-16 network, after training under the exact same setting (a setting we found to maximize a validation-set accuracy for the vanilla topologies).

|  |  | | learned | | | vanilla |
| --- | --- | --- | --- | --- | --- | --- |
|  |  | conv.5 | conv.7 | conv.10 | classifier | – |
| CIFAR 10 | accuracy | 90.6 | 92.61 | 92.94 | 92.79 | 92.32 |
|  | # parameters | 0.10 | 0.15 | 0.52 | 0.98 | 1.00 |
| CIFAR 100 | accuracy | 63.17 | 68.91 | 70.68 | 69.14 | 68.86 |
|  | # parameters | 0.10 | 0.13 | 0.52 | 0.98 | 1.00 |

Table 3: Classification accuracy (%) and overall network size (normalized number of parameters). VGG-16 is the "vanilla" topology. For both, CIFAR 10/100 benchmarks, the learned structure achieves the highest accuracy by replacing all the layers that are deeper than layer conv.10. Moreover, accuracy is maintained when replacing the layers deeper than layer conv.7.

One interesting phenomenon to note is that the highest accuracy is achieved at conv. 10 rather than at the "classifier" (the last dense layer). This might imply that although convolutional layers are useful at extracting features directly from images, they might be redundant for deeper layers. By using our structure learning algorithm to learn the deeper layers, accuracy of the overall structure increases with the benefit of having a compact network. An accuracy, similar to that of "vanilla" VGG-16, is achieved with a structure having 85% less total parameters (conv. 7) than the vanilla network, where the learned structure is over 50X smaller than the replaced part.

Next, we evaluate the accuracy of the learned structure as a function of the number of parameters and compare it to a densely connected network (fully connected layers) having the same depth and size. For SVHN, we used the Batch Normalized Maxout Network in Network topology (Chang & Chen, 2015) and removed the deepest layers starting from the output of the second NiN block (MMLP-2-2). For CIFAR-10, we used the VGG-16 and removed the deepest layers starting from the output of conv.10 layer. For MNIST, a structure was learned directly from pixels. Results are depicted in Figure 6. It is evident that accuracy of the learned structures is significantly higher (error bars represent 2 standard deviations) than a set of fully connected layers, especially in cases where the network is limited to a small number of parameters.

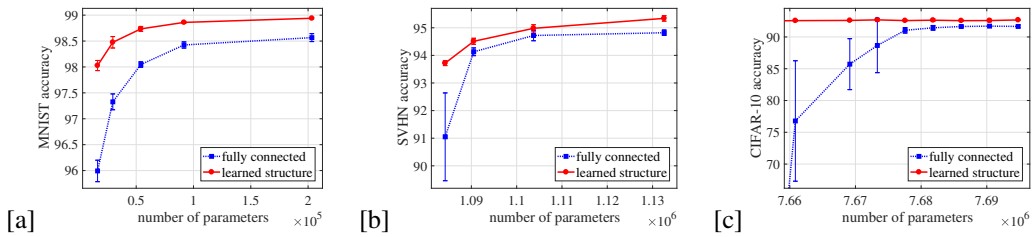

Figure 6: Accuracy as a function of network size. [a] MNIST, [b] SVHN. [c] CIFAR-10. Error bars represent 2 standard deviations.

Finally, in Table 4 we provide a summary of network sizes and classification accuracies, achieved by replacing the deepest layers of common topologies (vanilla) with a learned structure. In the first row, a structure is learned directly from images; therefore, it does not have a "vanilla" topology as reference (a network with 3 fully-connected layers having similar size achieves 98.75% accuracy). In all the cases, the size of the learned structure is significantly smaller than the vanilla topology, and generally has an increase in accuracy.

**Comparison to other methods**. Our structure learning algorithm runs efficiently on a standard desktop CPU, while providing structures with competitive classification accuracies and network sizes. For example, the lowest classification error rate achieved by our *unsupervised* algorithm for CIFAR 10 is 4.58% with a network of size 6M (WRN-40-4 row in Table 4). For comparison, the NAS algorithm (Zoph & Le, 2016) achieves error rates of 5.5% and 4.47% for networks of sizes 4.2M and 7.1M, respectively, and requires optimizing thousands of networks using hundreds of GPUs. For AlexNet network, recent methods for reducing the size of a pre-trained network (pruning while maintaining classification accuracy) achieve $5\times$ (Denton et al., 2014) and $9\times$ (Han et al., 2015; 2016) reduction. Our method achieves $13\times$ reduction. For VGG-16 (CIFAR-10), Li et al. (2017) achieve $3\times$ reduction and our method achieves $7\times$ reduction.

| dataset | topology | accuracy | | number of parameters | | |
| --- | --- | --- | --- | --- | --- | --- |
| | | vanilla | learned structure | "feature extraction" | removed section | learned structure |
| MNIST | None | | 99.07 (0.01) | | | |
| | MNIST-Man | 99.35 | 99.45 (0.04) | 23K | 104K | 24.7K |
| SVHN | Maxout NiN | 98.10 | 97.70 (0.05) | 1.07M | 527K | 52.6K |
| | SVHN-Man | 97.10 | 96.24 (0.05) | 17K | 88K | 25.5K |
| CIFAR 10 | VGG-16 | 92.32 | 92.61 (0.14) | 1.7M | 13.3M | 0.47M |
| | WRN-40-4 | 95.09 | 95.42 (0.14) | 4.3M | 4.7M | 1.7M |
| CIFAR 100 | VGG-16 | 68.86 | 68.91 (0.17) | 1.7M | 13.3M | 0.25M |
| ImageNet | AlexNet | 57.20 | 57.20 (0.03) | 2M | 59M | 2.57M |

Table 4: A summary of network sizes and classification accuracies (and standard deviations), achieved by replacing the deepest layers of common topologies (vanilla) with a learned structure. The number of parameters are reported for "feature extraction" (first layers of the vanilla topology), removed section (the deepest layers of the vanilla topology), and the learned structure that replaced the removed part. The sum of parameters in the "feature extraction" and removed parts equals to the vanilla topology size. The first row corresponds to learning a structure directly from image pixels.

## 6 CONCLUSIONS

We presented a principled approach for learning the structure of deep neural networks. Our proposed algorithm learns in an unsupervised manner and requires small computational cost. The resulting structures encode a hierarchy of independencies in the input distribution, where a node in one layer may connect another node in any deeper layer, and depth is determined automatically.

We demonstrated that our algorithm learns small structures, and maintains high classification accuracies for common image classification benchmarks. It is also demonstrated that while convolution layers are very useful at exploiting domain knowledge, such as spatial smoothness, translational invariance, and symmetry, they are mostly outperformed by a learned structure for the deeper layers. Moreover, while the use of common topologies (meta-architectures), for a variety of classification tasks is computationally inefficient, we would expect our approach to learn smaller and more accurate networks for each classification task, uniquely.

As only unlabeled data is required for learning the structure, we expect our approach to be practical for many domains, beyond image classification, such as knowledge discovery, and plan to explore the interpretability of the learned structures.

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

APPENDIX

## A  PRESERVATION OF CONDITIONAL DEPENDENCE

Conditional dependence relations encoded by the genrative structure $\mathcal{G}$ are preserved by the discriminative structure $\mathcal{G}_{\mathrm{D}}$ conditioned on the class $Y$. That is, $\mathcal{G}_{\mathrm{D}}$ conditioned on $Y$ can mimic $\mathcal{G}$; denoted by preference relation $\mathcal{G} \preceq \mathcal{G}_{\mathrm{D}}|Y$. While the parameters of a model can learn to mimic conditional independence relations that are not expressed by the graph structure, they are not able to learn conditional dependence relations (Pearl, 2009).

**Proposition 1.** *Graph $\mathcal{G}_{\mathrm{Inv}}$ preserves all conditional dependencies in $\mathcal{G}$ (i.e., $\mathcal{G} \preceq \mathcal{G}_{\mathrm{Inv}}$).*

*Proof.* Graph $\mathcal{G}_{\mathrm{Inv}}$ can be constructed using the procedures described by Stuhlmüller et al. (2013) where nodes are added, one-by-one, to $\mathcal{G}_{\mathrm{Inv}}$ in a reverse topological order (lowest first) and connected (as a child) to existing nodes in $\mathcal{G}_{\mathrm{Inv}}$ that d-separate it, according to $\mathcal{G}$, from the remainder of $\mathcal{G}_{\mathrm{Inv}}$. Paige & Wood (2016) showed that this method ensures the preservation of conditional dependence $\mathcal{G} \preceq \mathcal{G}_{\mathrm{Inv}}$. We set an equal topological order to every pair of latents $(H_i, H_j)$ sharing a common child in $\mathcal{G}$. Hence, jointly adding nodes $H_i$ and $H_j$ to $\mathcal{G}_{\mathrm{Inv}}$, connected by a bi-directional edge, requires connecting them (as children) only to their children and the parents of their children ($H_i$ and $H_j$ themselves, by definition) in $\mathcal{G}$. That is, without loss of generality, node $H_i$ is d-separated from the remainder of $\mathcal{G}_{\mathrm{Inv}}$ given its children in $\mathcal{G}$ and $H_j$. ∎

It is interesting to note that the stochastic inverse $\mathcal{G}_{\mathrm{Inv}}$, constructed without adding inter-layer connections, preserves all conditional dependencies in $\mathcal{G}$.

**Proposition 2.** *Graph $\mathcal{G}_{\mathrm{D}}$, conditioned on $Y$, preserves all conditional dependencies in $\mathcal{G}_{\mathrm{Inv}}$ (i.e., $\mathcal{G}_{\mathrm{Inv}} \preceq \mathcal{G}_{\mathrm{D}}|Y$).*

*Proof.* It is only required to prove that the dependency relations that are represented by bi-directional edges in $\mathcal{G}_{\mathrm{Inv}}$ are preserved in $\mathcal{G}_{\mathrm{D}}$. The proof follows directly from the d-separation criterion (Pearl, 2009). A latent pair $\{H, H'\} \subset \boldsymbol{H}^{(n+1)}$, connected by a bi-directional edge in $\mathcal{G}_{\mathrm{Inv}}$, cannot be d-separated by any set containing $Y$, as $Y$ is a descendant of a common child of $H$ and $H'$. In Algorithm 2-line 12, a latent in $\boldsymbol{H}^{(n)}$ is connected, as a child, to latents $\boldsymbol{H}^{(n+1)}$, and $Y$ to $\boldsymbol{H}^{(0)}$. ∎

We formulate $\mathcal{G}_{\mathrm{Inv}}$ as a projection of another latent model (Pearl, 2009) where bi-directional edges represent dependency relations induced by latent variables $\boldsymbol{Q}$. We construct a discriminative model by considering the effect of $\boldsymbol{Q}$ as an explaining-away relation induced by a class node $Y$. Thus, conditioned on $Y$, the discriminative graph $\mathcal{G}_{\mathrm{D}}$ preserves all conditional (and marginal) dependencies in $\mathcal{G}_{\mathrm{Inv}}$.

**Proposition 3.** *Graph $\mathcal{G}_{\mathrm{D}}$, conditioned on $Y$, preserves all conditional dependencies in $\mathcal{G}$ (i.e., $\mathcal{G} \preceq \mathcal{G}_{\mathrm{D}}$).*

*Proof.* It immediately follows from Propositions 1 & 2 that $\mathcal{G} \preceq \mathcal{G}_{\mathrm{Inv}} \preceq \mathcal{G}_{\mathrm{D}}$ conditioned on $Y$. ∎

Thus $\mathcal{G} \preceq \mathcal{G}_{\mathrm{Inv}} \preceq \mathcal{G}_{\mathrm{D}}$ conditioned on $Y$.

