# OpenReview forum: "Unsupervised Deep Structure Learning by Recursive Dependency Analysis"
_ICLR.cc/2018/Conference — Reject_

### Official Review · AnonReviewer1 · 2017-11-27
**There is a major technical flaw in this paper. And some experiment settings are not convincing.**

**Rating:** 4
**Confidence:** 4

**Review:**

The paper proposes an unsupervised structure learning method for deep neural networks. It first constructs a fully visible DAG by learning from data, and decomposes variables into autonomous sets. Then latent variables are introduced and stochastic inverse is generated. Later a deep neural network structure is constructed based on the discriminative graph. Both the problem considered in the paper and the proposed method look interesting. The resulting structure seems nice.

However, the reviewer indeed finds a major technical flaw in the paper. The foundation of the proposed method is on preserving the conditional dependencies in graph G. And each step mentioned in the paper, as it claims, can preserve all the conditional dependencies. However, in section 2.2, it seems that the stochastic inverse cannot. In Fig. 3(b), A and B are no longer dependent conditioned on {C,D,E} due to the v-structure induced in node H_A and H_B. Also in Fig. 3(c), if the reviewer understands correctly, the bidirectional edge between H_A and H_B is equivalent to H_A <- h -> H_B, which also induces a v-structure, blocking the dependency between A and B. Therefore, the very foundation of the proposed method is shattered. And the reviewer requests an explicit explanation of this issue.

Besides that, the reviewer also finds unfair comparisons in the experiments.

1. In section 5.1, although the authors show that the learned structure achieves 99.04%-99.07% compared with 98.4%-98.75% for fully connected layers, the comparisons are made by keeping the number of parameters similar in both cases. The comparisons are reasonable but not very convincing. Observing that the learned structures would be much sparser than the fully connected ones, it means that the number of neurons in the fully connected network is significantly smaller. Did the authors compare with fully connected network with similar number of neurons? In such case, which one is better? (Having fewer parameters is a plus, but in terms of accuracy the number of neurons really matters for fair comparison. In practice, we definitely would not use that small number of neurons in fully connected layers.)

2. In section 5.2, it is interesting to observe that using features from conv10 is better than that from last dense layer. But it is not a fair comparison with vanilla network. In vanilla VGG-16-D, there are 3 more conv layers and 3 more fully connected layers. If you find that taking features from conv10 is good for the learned structure, then maybe it will also be good by taking features from conv10 and then apply 2-3 fully-connected layers directly (The proposed structure learning is not comparable to convolutional layers, and what it should really compare to is fully-connected layers.) In such case, which one is better?
Secondly, VGG-16 is a large network designed for ImageNet data. For small dataset such as CIFAR10 and CIFAR100, it is really overkilled. That's maybe the reason why taking the output of shallow layers could achieve pretty good results.

3. In Fig. 6, again, comparing the learned structure with fully-connected network by keeping parameters to be similar and resulting in large difference of the number of neurons is unfair from my point of view.

Furthermore, all the comparisons are made with respect to fully-connected network or vanilla CNNs. No other structure learning methods are compared with. Reasonable baseline methods should be included.

In conclusion, due to the above issues both in method and experiments, the reviewer thinks that this paper is not ready for publication.

---

> ### Author Response · Authors · 2017-12-08
> **Clarifications regarding the technical flaw (and more)**
>
> We'd like to thank the reviewer for the feedback.
> As for the points that were raised:
>
> ---------------------------------
> Reviewer's point:
> >> the reviewer indeed finds a major technical flaw in the paper....
>
> Our response:
> Our method is indeed based on the preservation of the conditional dependencies, encoded by the generative model, in the discriminative model. The reviewer's observation regarding the disability of the inverse model to preserve conditional dependencies among the observed variable is correct, as indeed apparent in figure (3).
> *However*, an important observation that we should have made clear, is that since we’re interested in learning a discriminative model that infers the latent variables (h) given the observed (A,B,C…) , the only relevant conditional dependencies that must be preserved are those among the hidden variables, and between the hidden variables and the observed. The conditional dependencies among the observed variables are not relevant to the inference of the hidden variable given the observed, thus its preservation is not handled during the model inversion. In the paper's appendix, we elaborate on the theoretical correctness of the inversion process and refer to Paige & Wood (2016) that proves the validity of the method. Specifically, refer to Figure (1) in Paige & Wood (2016). If we were interested in preserving the conditional dependencies among the observed variables, we would have ended up with the middle (b) structure. Since we're not interested in those dependencies, we proceed and eventually end up with the right structure (c)
> *We’d like to thank the reviewer for pointing out this important observation. We’ll edit the paper to better clarify it.*
>
>
> ------------------------------------
> Reviewer's point:
> >>  In section 5.1, although the authors show that the learned structure achieves 99.04%-99.07% compared with 98.4%-98.75% for fully connected layers, the comparisons are made by keeping the number of parameters similar in both cases....
>
>
> Our response:
> In our evaluation we followed the common reporting protocol we have encountered when surveying the literature on the topic. Other papers used the number of parameters as a measure of network capacity (e.g. Real et al 2017, Zoph et al 2017).
> Having said that, referring to figure 6 we can see that increasing the number of neurons in the fully connected layers, the accuracy eventually converges to a limit that is significantly (statistically-wise) below the learned structure.
>
> -----------------------------------------------
> Reviewer's point:
> >>In section 5.2, .... If you find that taking features from conv10 is good for the learned structure, then maybe it will also be good by taking features from conv10 and then apply 2-3 fully-connected layers directly...
> Secondly, VGG-16 is a large network designed for ImageNet data. For small dataset such as CIFAR10 and CIFAR100, it is really overkilled. That's maybe the reason why taking the output of shallow layers could achieve pretty good results
>
>
> Our response:
> As for the 1st comment regarding removal of the last conv layers from VGG - that's a correct observation. Please refer to figure 6 in our paper, that describes the result of such experiment on the three datasets we used for evaluation.
>
> As for the 2nd comment regarding VGG being overkill for CIFAR - Note that the VGG we used for CIFAR is much smaller than the version used for ImageNet (15M vs 130M parameters). In addition, in our experiments we modified the network’s size per each dataset where it made sense (SVHN and CIFAR-10) in order to increase our coverage. VGG was chosen for CIFAR-10 as it is commonly used as a reference structure in the relevant literature and appears in the CIFAR-10 results leader-board.
> We had also conducted an experiment (see table 3 in the paper) in which we cut VGG at different layers and measured the accuracy. We observed graceful reduction in accuracy as we removed conv layers.

---

### Official Review · AnonReviewer2 · 2017-11-27
**Interesting unsupervised structure learning algorithm**

**Rating:** 5
**Confidence:** 2

**Review:**

This paper tackles the important problem of structure learning by introducing an unsupervised algorithm, which encodes a hierarchy of independencies in the input distribution and allows introducing skip connections among neurons in different layers. The quality of the learnt structure is evaluated in the context of image classification, analyzing the impact of the number of parameters and layers on the performance.

The presentation of the paper could be improved. Moreover, the paper largely exceeds the recommended page limit (11 pages without references).

My main comments are related to the experimental section:

- Section 5 highlights that experiments were repeated 5 times; however, the standard deviation of the results is only reported for some cases. It would be beneficial to include the standard deviations of all experiments in the tables summarizing the obtained results.

- Are the differences among results presented in table 1 (MNIST) and table 2 (CIFAR10) statistically significant?

- It is not clear how the numbers of table 4 were computed (size replaced, size total, t-size, replaced-size). Would it be possible to provide the number of parameters of the vanilla model, the pre-trained feature extractor and the learned structure separately?

- In section 5.2., there is only one sentence mentioning comparisons to alternative approaches. It might be worth expanding this and including numerical comparisons.

- It seems that the main focus of the experiments is to highlight the parameter reduction achieved by the proposed algorithm. There is a vast literature on model compression, which might be worth reviewing, especially given that all the experiments are performed on standard image classification tasks.

---

> ### Author Response · Authors · 2017-12-08
> **Further clarifications**
>
> We'd like to thank the reviewer for the feedback.
> As for the points that were raised:
> ------------------------
> Reviewer's points:
> >> It would be beneficial to include the standard deviations of all experiments in the tables summarizing the obtained results.
> >> Are the differences among results presented in table 1 (MNIST) and table 2 (CIFAR10) statistically significant?
>
> Our response:
> As indicated in the paper, we have recorded the standard deviation and will edit the paper to add this missing data that will prove the significance of the differences.
>
> -----------------------
> Reviewer's point:
> >>  It is not clear how the numbers of table 4 were computed (size replaced, size total, t-size, replaced-size). Would it be possible to provide the number of parameters of the vanilla model, the pre-trained feature extractor and the learned structure separately?
>
> Our response:
> First, thanks for the comment. we will update the table to be clearer and easier to digest.
>
> We'll walk through one example from that table and clarify, hoping this will help understanding the rest of the lines:
> Lets refer to line #2 , analyzing the "MNIST-Man" topology. The vanilla topology was manually constructed (thus the name MNIST-Man), its total size is 127K parameters and it achieve a classification accuracy of 99.35%
> If we look at the network as composed of a 'head' (from the first conv layer up to a certain layer at the depth of the network) and a 'tail' (all the subsequent layers up to the softmax layer), then what we did is to throw out the 'tail' and replace it with a learned structure (similar to what is done in transfer learning, only here the 'tail' is much larger). In this specific case, the tail whose size is 104K parameters in the original network was replaced by a learned structure whose size is 24% (0.24 - taken from the 'replaced size' column) of the 107K (i.e. ~26K parameters) which reflects a reduction of 4.2X in tail size.
> The overall size of the new network (original head + learned tail) is therefore 49K (=23K (head size) + 26K (tail size)) 49K which is 38% (0.38='t-size') of the vanilla network size (127K) and achieves a classification accuracy of 99.45%
>
> ------------------------------------
> Reviewer's point:
> >> In section 5.2., there is only one sentence mentioning comparisons to alternative approaches. It might be worth expanding this and including numerical comparisons.
>
> Our response:
> In section 5.2 we have mentioned a comparison to one of the prominent papers (at the time of submission ) indicating the following:
> - On CIFAR-10 our method, based on Wide Resnet topology has achieved 95.42% accuracy with network size of 6M parameters
> - NAS (Zoph et al 2016) have achieved 94.5% and 95.53% accuracy for networks of sizes 4.2M and 7.1M respectively
> - DeepArchitect (Negrinho et al 2017) report 89% accuracy on CIFAR-10 . They havent provided any further details to conduct elaborate comparison
> - Baker et al. achieved 93.08% accuracy on a single top-performing model
> - Real et al. achieved 94.6% accuracy with network size of 5.4M parameters
> - Other related works (Smithson et al. , Miikkulainen et al.)  have not tested on CIFAR-10 or used significantly different metric thus not easily comparable to our methods
>
> It’s interesting to note that none of the other papers have provided statistical significance measures (std dev) in their results and represented the capacity of the network by its size - measured by number of parameters.

---

### Official Review · AnonReviewer4 · 2017-12-07
**Promising method, inconclusive results**

**Rating:** 5
**Confidence:** 3

**Review:**

Authors propose a deep architecture learning algorithm in an unsupervised fashion. By finding conditional in-dependencies in input as a Bayesian network and using a stochastic inverse mechanism that preserves the conditional dependencies, they suggest an optimal structure of fully connected hidden layers (depth, number of groups and connectivity). Their algorithm can be applied recursively, resulting in multiple layers of connectivity. The width of each layer (determined by number of neurons in each group) is still tuned as a hyper-parameter.

Pros:
- Sound derivation for the method.
- Unsupervised and fast algorithm.
Cons:
- Poor writing, close to a first draft.
- Vague claims of the gain in replacing FC with these structures, lack of comparison with methods targeting that claim.
 - If the boldest claim is to have a smaller network, compare results with other compression methods.
 - If it is the gain in accuracy compare with other learn to learn methods and show that you achieve same or higher accuracy. The NAS algorithm achieves 3.65% test error. With a smaller network than the proposed learned structure (4.2M vs 6M) here they achieve slightly worse (5.5% vs 4.58%) but with a slightly larger (7.1M vs 6M) they achieve slightly better results (4.47% vs 4.58%). The winner will not be clear unless the experiments fixes one of variables or wins at both of them simultaneously.

Detailed comments:

- Results in Table 4 mainly shows that replacing fully connected layer with the learned structures leads to a much sparser connectivity (smaller number of parameters) without any loss of accuracy. Fewer number of parameters usually is appealing either because of better generalizability or less computation cost. In terms of generalizability, on most of the datasets the accuracy gain from the replacement is not statistically significant. Specially without reporting the standard deviation. Also the generalizability impact of this method on the state-of-the-art is not clear due to the fact that the vanilla networks used in the experiments are generally not the state-of-the-art networks. Therefore, it would be beneficial if the authors could show the speed impact of replacing FC layers with the learned structures. Are they faster to compute or slower?
- The purpose of section 5.1 is written as number of layers and number of parameters. But it compares with an FC network which has same number of neurons-per-layer. The rest of the paper is also about number of parameters. Therefore, the experiments in this section should be in terms of number of parameters as well. Also most of the numbers in table 1 are not significantly different.

Suggestions for increasing the impact:

This method is easily adaptable for convolutional layers as well. Each convolutional kernel is a fully connected layer on top of a patch of an image. Therefore, the input data rather than being the whole image would be all patches of all images. This method could be used to learn a new structure to replace the KxK fully connected transformation in the convolutional layer.

The fact that this is an unsupervised algorithm and it is suitable for replacing FC layers suggests experimentation on semi-supervised tasks or tasks that current state-of-the-art relies more on FC layers than image classification. However, the experiments in this paper are on fully-labeled image classification datasets which is possibly not a good candidate to verify the full potential of this algorithm.

---

> ### Author Response · Authors · 2017-12-21
> **Clarifications**
>
> We would like to thank the reviewer for the thorough review and important suggestions.
>
> reviewer:
> >> “Promising method, inconclusive results”
>
> Our response:
> We significantly improved the clarity of our main experimental results (Table 4). We reordered the columns and report the absolute sizes of: “feature extraction”, replaced size, and learned structure size. We also included, as requested, a comparison to recent compression (pruning) methods.
> -----------------------
>
> reviewer:
> >> "If the boldest claim is to have a smaller network, compare results with other compression methods."
>
> Our response:
> We added a comparison to recent model compression (pruning) methods. In all the compared cases, our algorithm learns smaller networks while preserving accuracy. Compression methods are commonly supervised and aimed at reducing the network size. In contrast to pruning methods that prune all the layer, our algorithm keeps the first few layers (“feature extraction”) of the network intact and removes the deeper layers altogether. It then learns a new deep structure in an unsupervised manner (and is very fast; Matlab implementation on CPU). The total size of the network, feature extraction+learned structure is smaller than that of networks resulting from pruning methods.
>
>  Compared to the NAS algorithm, our method is on par in terms of accuracy and model size but significantly faster.
> -----------------------
>
> reviewer:
> >> "The purpose of section 5.1 is written as number of layers and number of parameters. But it compares with an FC network which has same number of neurons-per-layer."
>
> Our response:
> The comparison to FC networks is where both networks have similar number of parameters and not number of neurons. We corrected and clarified this in the paper.
> -----------------------
>
> reviewer: (Suggestions for increasing the impact 1)
> >> "This method is easily adaptable for convolutional layers as well."
>
> Our response:
> This is a very important idea. In fact, we have been working on it for the past several months. However, there are additional concepts that need to be introduced, which we believe will complicate the current paper. For example, it is required to derive a new conditional independence test for comparing two patches conditioned on a set of other patches. Moreover, the varying size of the receptive field, as a function of the network depth, should be accounted for in the test (recall that only the input image pixels are considered in all the independence tests).
> In this paper, we lay the foundations of the algorithm and demonstrate its effectiveness by learning the "classifier" part of the network (replacing the deepest of both convolutional and FC layers).
> -----------------------
>
> Reviewer: (Suggestions for increasing the impact 2)
> >> The fact that this is an unsupervised algorithm ... However, the experiments in this paper ... possibly not a good candidate to verify the full potential of this algorithm."
>
> Our response:
> We agree that the experiments do not cover the full potential of the algorithm; however, the experiments in this paper demonstrate the key idea of learning an efficient structure from unlabeled data. Efficiency is demonstrated by learning structures that are significantly smaller (while retaining classification accuracy or improving it) than a stack of convolutional and FC layers, as used in a range of common topologies.

---

### Decision · Program_Chairs · 2018-01-29
**ICLR 2018 Conference Acceptance Decision**

**Decision:**

Reject

**Comment:**

The updated draft has helped to address some of the issues that the reviewers had, however the reviewers believe there are still outstanding issues. With regard to the technical flaw, one reviewer has pointed out that the update changes the story of the paper by breaking the connection between the generative and discriminative model in terms of preserving or ignoring conditional dependencies.

In terms of the experiments, the paper has been improved by the reporting of standard deviation, and comparison to other works. However it is recommended that the authors compare to NAS by fixing the number of parameters and reporting the results to facilitate an apples-to-apples comparison. Another reviewer also recommends comparing to other architectures for a fixed number of neurons.